# Biogas and Methane Potential of Pre-Thermally Disintegrated Bio-Waste

**Sylwia Myszograj** 

Faculty of Civil Engineering, Architecture and Environmental Engineering, Institute of Environmental Engineering, University of Zielona Góra, 65-516 Zielona Góra, Poland; S.Myszograj@iis.uz.zgora.pl; Tel.: +48-6832-825-74

**Abstract:** One of the environmental solutions employed in order to achieve circular economy goals is methane fermentation—a technology that is beneficial both for the stabilization and reduction of organic waste and for alternative energy generation. The article presents the results of research aimed at determining the biogas and methane potential of bio-waste which has been pre-thermally disintegrated, and determining the influence of variable process parameters of disintegration on the kinetics of fermentation. A first-order kinetic model was used to describe the fermentation as well as two mathematical models: logistic and Gompertz. It has been found that process parameters such as time (0.5, 1 and 2 h) and temperature (between 55 to 175 °C) have a significant effect on the solubilization efficiency of the bio-waste. The methane fermentation of thermally disintegrated bio-waste showed that the highest biogas potential is characterized by samples treated, respectively, for 0.5 h at 155 °C and for 2 h at 175 °C. The best match for the experimental data of biogas production from disintegrated substrates was demonstrated for the Gompertz model.

**Keywords:** biogas; biowaste; kinetics

## 1. Introduction

Achieving economic growth while respecting the environment is key to sustainable development, which is pursued by rationally managing natural resources, improving energy efficiency, and reducing greenhouse gas emissions. European Union waste management regulations are based on three directives: 94/62/EC on packaging and packaging waste [1], 1999/31/EC on the landfill of waste [2], and 2008/98/EC on waste [3]. The EU aims to create a "recycling society" aimed at "avoiding waste generation and using waste as a resource". Existing restrictions on the storage of biodegradable waste necessitate either biological or thermal processing methods. This type of waste is characterized by high moisture, which makes biological processes more conducive to its treatment than thermal ones. Biological methods may be carried out under aerobic conditions (aerobic stabilization or composting), anaerobic conditions (methane fermentation), or a combination of both technologies. The fermentation of organic waste best meets binding legal and processing requirements because of the following factors [4–6]:

- It is a way to recycle separately collected organic waste, which is processed into high-quality fertilizer that improves soil fertility;
- There is a possibility of energy recovery (biogas generation);
- The volume of the organic waste fraction is reduced by more than 30%, depending on the duration of the fermentation process, whereas the landfilled products can be concentrated in up to 1.3 Mg/m$^3$, which allows optimal use of available landfill capacity;

- The fermentation process does not pose a risk of creating toxic chemical compounds, whereas organic substances present in waste can be partially transformed through the life processes of micro-organisms;
- It limits the intensity of processes that later occur in landfilled waste. Biologically processed waste consists to a large extent of an inert fraction, which is the reason that, when landfilled, it emits much smaller amounts of biogas and leachate with low pollutant concentrations.

Methane fermentation is a multistage process, with the main phases being hydrolysis, acidogenesis, acetogenesis, and methanogenesis. It is known that the phase limiting the rate of anaerobic decomposition processes of substrates with high solid matter content is, among others, the rate of enzymatic hydrolysis (liquefaction) of insoluble organic polymers to soluble forms available for microorganisms [7,8]. Operational and research experience shows that the potential benefits of this method of waste treatment prior to methane fermentation include the increased biodegradability of substrates, reduction and improvement of fermentate drainage and hygienization, lower odor emissions during stabilization, higher biogas production, and an enhanced energy balance of the process [9–11].

Regardless of the disintegration method used, the primary effect of the treatment is the liquefaction of the substrate's solid fraction. It is assumed that the fragmentation of sludge or solid waste will increase the availability of organic substances for micro-organisms; i.e., the potential biodegradability [12]. However, the existing literature does not confirm this fact unequivocally. According to research by Wang et al. [13], the increased biodegradability of substrates is linked to their liquefaction and particle size reduction, but other researchers show no correlation between these parameters [14,15]. Depending on the characteristics of the raw substrates and the processing method, the increase in the biodegradability of hydrolysates may be limited by the formation of refractory/toxic compounds [16,17] and by the decomposition (loss) of organic matter [18]. Biodegradation inhibitors arise from, among others, the disintegration of biomass containing lignocellulose, which results in the creation of hydroxymethylfurfural (HMF) and soluble phenol compounds [18] or Amadori and melanoid products (Maillard reaction products) [19,20].

Assessing the biodegradability of hydrolysates by determining the biogas and methane potential is the most practical of all indicators for establishing the effectiveness of disintegration methods. Estimating the amount of obtainable biogas directly indicates an improvement of the energy effect on the volume or dry matter of the disintegrated substrates. Simple methods for determining biogas potential are based on formulas by Baserg [21], Keymer and Schlicher [22], and Amon et al. [23]. They were developed independently in different years and are founded on basic organic substrates, while the efficiency of the methane fermentation process is based on an assessment of the production of methane and carbon dioxide.

One of the most commonly used equations in modeling the process of methane fermentation is the Gompertz model [24]. This formula was the basis for the description of research carried out by Nopharatana et al. [25], Altas [26], Zhu et al. [27], Budiyono et al. [28], and Patil et al. [29]. The Gompertz equation can be used to describe the growth rate of micro-organisms. The solution to this equation is a sigmoid curve of biomass growth, whereas the formula contains mathematical parameters of no biological consequence. Zwietering et al. [24] modified the Gompertz equation by including parameters such as the maximum amount of biomass, maximum population growth rate, and lag phase. At present, many authors use the modified Gompertz model to determine the rate of biogas production, assuming its correlation with the rate of anaerobic biomass growth.

This paper presents the results of research aimed at determining the biogas and methane potential of bio-waste disintegrated thermally and the influence of variable process parameters of disintegration on the kinetics of methane fermentation. It should be stressed that the use of bio-waste as a substrate was a significant part of the research. Publications showing research results for this substrate are scarce, while those that are available are mostly concerned with selected temperatures and process times of food industry and agriculture waste. Information on the disintegration of municipal waste concerns

only some of the disintegration methods due to the very diverse morphological and physico-chemical composition of this substrate.

## 2. Materials and Methods

Municipal waste collected from an area of new multi-family buildings was used for the research. The determination of the granular composition consisted of sieving the waste and weighing the obtained 0–10 mm, 10–20 mm, and 20–80 mm fractions. Oversized fractions, measuring over 80 mm, were rejected. The research material, therefore, consisted of biodegradable components separated from the 0–80 mm fraction. For this purpose, glass, paper, plastics, metals, textiles, mineral wastes, and other non-bio-waste were separated from waste. The remaining waste was divided into characteristic components and their share in the total sample weight was determined. The morphological composition of bio-waste used in the study is presented in Table 1.

**Table 1.** Morphological composition of bio-waste used in the research; average values.

| Component | | Percentage, % |
|---|---|---|
| 10–20 mm fraction | | 26.0 |
| Wood | | 1.0 |
| Potato peelings | | 25.7 |
| Carrot peelings | | 4.9 |
| Banana peel | | 7.3 |
| Tangerine and orange peelings | | 4.0 |
| Lemons—peelings, slices from tea | | 2.5 |
| Onion and leek leftovers | | 1.3 |
| Cabbage leaves | | 4.6 |
| Meat, cold cuts | 20-80 mm kitchen waste | 3.2 |
| Fish bones and skin | | 1.0 |
| Poultry carcasses | | 1.2 |
| Other bones | | 5.0 |
| Boiled pasta | | 3.7 |
| Bread | | 3.7 |
| Egg shells | | 1.7 |
| Teabags | | 2.7 |
| Sunflower husks | | 0.5 |

(20-80 mm kitchen waste total: 73.0)

In each series of tests, fresh raw waste was prepared according to morphological composition and crushed into a granular fraction not exceeding 20 mm. Water extracts were prepared for the examined bio-waste and analyzed for physical and chemical composition. The tested sample consisted of 50 g substrate (approx. 30 g DM) and 200 $cm^3$ of distilled water. The physical and chemical characteristics of the bio-waste and prepared water extracts are presented in Tables 2 and 3, respectively.

**Table 2.** Physicochemical properties of the bio-waste used in the study.

| Parameter | Unit | Values | Average Value ± Dev. Stand. |
|---|---|---|---|
| Moisture | % | 54.5–61.8 | 59.8 ± 1.1 |
| Loss on ignition | % DM | 62.2–71.5 | 64.0 ± 0.5 |
| Total organic carbon | % DM | 24.4–32.9 | 29.4 ± 0.7 |
| Total Kiejdahl nitrogen | % DM | 1.00–1.40 | 1.10 ± 0.05 |
| Chemical Oxygen Demand | mg $O_2$/g DM | 934–978 | 951 ± 16 |

**Table 3.** Physicochemical properties of water extracts from bio-waste used in the study.

| Parameter | Unit | Values | Average Value ± Dev. Stand. |
|---|---|---|---|
| pH | - | 7.2–7.4 | - |
| Dry mass | mg DM/L | 16000–18900 | 17100 ± 688 |
| Loss on ignition | mg DOM/L | 1630–5890 | 3180 ± 1790 |
| Total organic carbon | mg C/L | 83–229 | 190 ± 42 |
| Total Kiejdahl nitrogen | mg N/L | 169–211 | 186 ± 10 |
| Chemical Oxygen Demand | mg O$_2$/L | 365–458 | 404 ± 26 |
| Volatile Fatty Acids | mg CH$_3$COOH/L | 60–85 | 71 ± 7 |

In the experiments, bio-waste was treated at temperatures of 55 °C, 75 °C, 95 °C, 115 °C, 135 °C, 155 °C, and 175 °C over a period of 0.5, 1, and 2 hours. Disintegration tests were performed in each of the process conditions, in three repetitions. Thermal disintegration was carried out in autoclave Zipperclave 1.0 L, manufactured by Autoclave Engineers. The autoclave equipment allows the control of temperature and treatment time as well as pressure regulation in the reactor. The autoclave heats up to the selected temperature within 10–20 min, while its cooling with cooling liquid, depending on the temperature of processing, lasts up to 15 min. The process of thermal disintegration was carried out without stirring.

For samples obtained in the disintegration process across the full range of tested temperatures and times, an analysis of the efficiency of mesophilic methane fermentation at 37 °C was carried out. The biogas and methane potential of the tested substrates in both raw form and after thermal disintegration was determined on a laboratory scale, at a stand with 12 bioreactors with a capacity of 1 L for batch fermentation with stirring. The reactors were filled with properly prepared samples, fermented sludge for grafting, and placed in a bathtub filled with water (thermostat set at 37 °C). After mixing the content and removing the air, the reactors were tightly connected with gas burettes with a capacity of 2.3 L to measure the volume of generated biogas. The burettes were filled with a saturated NaCl solution which had been stored in equalizing tanks.

In the fermentates, the pH, redox potential, and alkalinity were monitored for general treatment control. During the process (28 d), the daily volume of produced gas and its composition (CH$_4$, CO$_2$, NH$_3$, H$_2$S) were measured with the use of the GA 2000 plus Geotechnical Inst gas analyzer. The volume of biogas was adjusted to standard temperature and pressure conditions. The results were recorded in accordance with standards and procedures in force at the Accredited Laboratory of the Environmental Engineering Institute of the University of Zielona Góra.

## 3. Description of the Mesophilic Methane Fermentation Process Kinetics

Biogas yield was estimated on the basis of first-order kinetic and mathematical models:
a. Model based on the first-order equation (FOM) [30]:

$$B_{(t)} = B_{max}(1 - \exp(-k_h \cdot t)) \tag{1}$$

where

B$_{[t]}$ is the cumulative biogas production during the duration *t* of fermentation, L/kg DOM;
B$_{max}$ is the maximum biogas yield, L/kg DOM;
k$_h$ is the kinetic rate constant, d$^{-1}$;
t is the duration of the process, d.

b. Modified logistic model (LM) [31]:

$$B_{(t)} = \frac{B_{max}}{1 + \exp(4R_m(\lambda - t)/B_{max} + 2)} \tag{2}$$

where

$R_m$ is the maximum rate of biogas production, L/(kg DOM·d);
$\lambda$ is the duration of the lag phase, d.

c. Modified Gompertz model (GM), [32]:

$$B_{(t)} = B_{max} \exp\left(-\exp\left(\frac{R_m \cdot e}{B_{max}}(\lambda - t) + 1\right)\right) \qquad (3)$$

Where e is the mathematical constant, 2.718.

The Akaike Information Criterion (AIC) and the Residual Sum of Squares (RSS) were used to verify the model estimation.

## 4. Results

### 4.1. Thermal Disintegration of Bio-Waste

The average concentration of solid matter in bio-waste amounted to 83.82 ± 2.09 g/L. Organic substances constituted, on average, 71% of DM (ranging from 65 to 77% DM), and the COD/DOM (Chemical Oxygen Demand/ Dry Organic Matter) quotient varied from 0.97 to 1.16 (median 1.05). Changes in the concentrations of solids and substances dissolved in bio-waste, depending on the temperature and duration of disintegration, are presented in Figure 1 with a division into mineral and organic fractions.

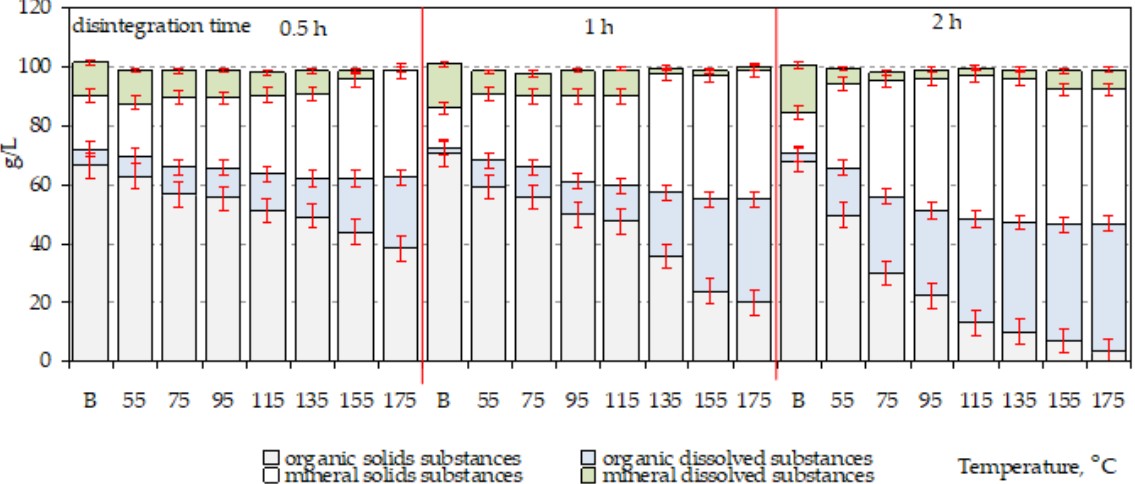

**Figure 1.** Changes in the concentration of the total suspension and in substances dissolved in bio-waste and hydrolysates, depending on the temperature and duration of disintegration.

The concentration of dissolved organic and solid mineral substances increased with the rise in temperature and processing time of bio-waste. The highest efficiency of organic solid waste fraction liquefaction was obtained during a disintegration lasting two hours. At 55 °C, the concentration of organic substances dissolved in hydrolysates was already similar to that obtained in the process lasting 0.5 and 1 h at 135 °C.

The content of mineral solids increased from an average value of 15.26 g/L in raw bio-waste to maximum values in samples disintegrated at 175 °C: 35.85 g/L (0.5 h), 43.84 g/L (1 h), and 45.35 g/L (2 h), respectively. In each series of tests, the dry residue in the disintegrated samples decreased on average by about 2%.

### 4.2. Biogas and Methane Potential of Raw and Thermally Disintegrated Bio-Waste

The total biogas and methane production capacity per kg of dry matter and kg of organic dry matter is presented in Table 4. The biogas potential of untreated bio-waste was 206 L /kg DOM, whereas the cumulative methane production amounted to 97 L CH4/kg DOM, which corresponded to a 47% share of this component in biogas. In thermally disintegrated samples, the amount of biogas obtained, and the share of methane, varied depending on the conditions of the liquefaction process.

The thermal disintegration of bio-waste over 0.5 h resulted in an increase in biogas production compared to the anaerobic stabilization of non-liquefied waste, ranging from 8–51%. The highest biogas and methane yield was obtained after the treatment of bio-waste at 155 °C (312 L/kg DOM and 201 L CH4/kg DOM). For samples liquefied at low temperatures, the share of methane stood at about 57%. High-temperature disintegration, apart from liquefaction at 175 °C, led to biogas production with a higher proportion of methane (65% at 155 °C). The highest daily yield of biogas was recorded on the fifth day of processing in a sample subjected to disintegration at 155 °C (33 L/(kg DOM·d)).

**Table 4.** Biogas and methane potential in raw and thermally disintegrated bio-waste.

| Time, h | Temperature, °C | Biogas Potential | | Methane Potential | | Methane Share in Biogas |
|---|---|---|---|---|---|---|
| | | L/kg DM | L/kg DOM | L/kg DM | L/kg DOM | % |
| Raw bio-waste | | 146 | 206 | 69 | 97 | 47 |
| 0.5 | 55 | 157 | 222 | 91 | 129 | 58 |
| | 75 | 151 | 225 | 86 | 129 | 57 |
| | 95 | 170 | 256 | 95 | 143 | 56 |
| | 115 | 191 | 294 | 116 | 178 | 61 |
| | 135 | 194 | 308 | 123 | 195 | 63 |
| | 155 | 197 | 312 | 127 | 201 | 65 |
| | 175 | 185 | 292 | 110 | 174 | 59 |
| 1 | 55 | 154 | 221 | 87 | 124 | 56 |
| | 75 | 158 | 234 | 92 | 136 | 58 |
| | 95 | 179 | 290 | 105 | 169 | 58 |
| | 115 | 188 | 309 | 117 | 193 | 62 |
| | 135 | 200 | 347 | 130 | 224 | 65 |
| | 155 | 200 | 357 | 133 | 238 | 67 |
| | 175 | 214 | 389 | 145 | 263 | 68 |
| 2 | 55 | 181 | 273 | 90 | 135 | 50 |
| | 75 | 176 | 310 | 89 | 157 | 51 |
| | 95 | 182 | 351 | 106 | 204 | 58 |
| | 115 | 185 | 383 | 125 | 259 | 68 |
| | 135 | 217 | 456 | 147 | 308 | 68 |
| | 155 | 245 | 519 | 155 | 327 | 63 |
| | 175 | 260 | 550 | 164 | 347 | 63 |

The total biogas production in samples that were thermally disintegrated in 1 h rose with the increase in processing temperature and ranged from 221–389 L/kg DOM. The share of methane in the biogas varied from 56–68%. Biogas production from liquefied waste exceeded the biogas potential of raw waste by 8–89%. The highest daily biogas yields were obtained on the fourth and fifth day of treatment, with the values ranging from 27–39 L/(kg DOM·d).

Thermal disintegration over a period of 2 h had the most beneficial effect on the biogas and methane potential of the bio-waste. The biogas yield ranged from 273–550 L/kg DOM and the methane yield from 135–347 L CH4/kg DOM. The share of methane in biogas depended on the disintegration temperature. In terms of low-temperature processing, it ranged from 50–58%, and was higher (68%) for disintegration temperatures of 115 °C and 135 °C; after treatment at temperatures of 155 °C and 175 °C, it decreased to 63%. Compared to the biogas potential of raw waste, biogas production was higher by 167% (175 °C). The highest daily biogas yield (47 L/(kg DOM·d)) was observed on the third day of methane fermentation in the samples subjected to high-temperature thermal disintegration (175 °C).

## 5. Discussion

Most studies on disintegration methods described in the literature are limited to the evaluation of process efficiency based on the COD liquefaction rate and the change in total biogas yield compared to the biogas potential of the raw substrate [33–35]. From a practical point of view, the aim of preliminary processing is not only to improve productivity but also to boost the speed of biogas production and the proportion of methane in it.

The impact of preliminary thermal disintegration on the process of methane fermentation of bio-waste was assessed by calculating the process efficiency parameters using the first-order kinetic model (FOM) and simplified mathematical models: the logistic model (LM) and the modified Gompertz equation (GM).

Figure 2 presents curves of models matching experimental data of total biogas yield from untreated bio-waste. Table 5 presents the values of estimated parameters for three analyzed models of methane fermentation for raw and thermally disintegrated bio-waste.

It was found that all models describe the experimental data correctly. The best match of the calculated curves of the mathematical models to the experimental data was obtained for the Gompertz model, which is also confirmed by observations made by Atlas [26] and Donoso-Bravo et al. [31]. The lowest residual standard error (RSE) and Akaike Information Criterion (AIC) values were obtained for this model. It is important to note the shape of the model curves. For the GM and LM models, when t = 0, biogas production is not equal to zero, which, of course, does not make any physical sense. A similar doubt was raised by Donoso-Bravo et al. [31].

For raw bio-waste, the highest value of the $B_{max}$ parameter stood at 219.7 L/kg DOM. The value of maximum $B_{max}$ biogas production for the disintegrated substrate was the highest for the first-order model and the lowest for the logistic model. It should be noted that, for most of the samples, the differences in the values determined in the three models do not exceed 5%.

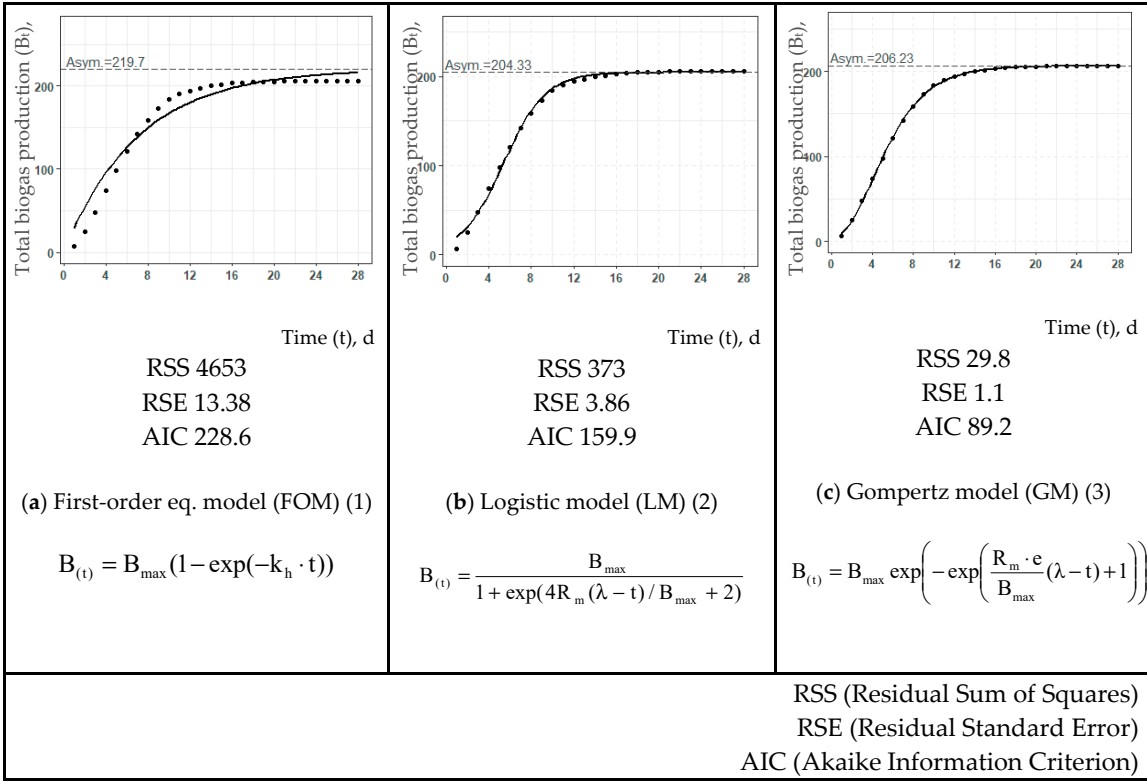

**Figure 2.** Curves of models matching experimental data of total biogas production (L/kg DOM) from raw bio-waste.

**Table 5.** Estimated values of parameters for three analyzed methane fermentation models for raw and thermally disintegrated bio-waste.

| T, °C | Estimated Values | First-Order eq. Model (FOM) (1) | | | Logistic Model (LM) (2) | | | Gompertz Model (GM) (3) | | |
|---|---|---|---|---|---|---|---|---|---|---|
| | | **Raw bio-waste** | | | | | | | | |
| **T, °C** | $B_{max}$, L/kg DOM | 220 | | | 204 | | | 206 | | |
| | $R_m$, L/(kg DOM·d) | - | | | 25.8 | | | 26.8 | | |
| | $k_h$, $d^{-1}$ | 0.142 | | | - | | | - | | |
| | λ, d | - | | | 1.41 | | | 1.30 | | |
| | | **Thermally disintegrated bio-waste—time of process, h** | | | | | | | | |
| | | 0.5 | 1 | 2 | 0.5 | 1 | 2 | 0.5 | 1 | 2 |
| 55 | $B_{max}$, L/kg DOM | 239 | 233 | 302 | 217 | 216 | 269 | 220 | 218 | 273 |
| | $R_m$, L/(kg DOM·d) | - | - | - | 21.2 | 22.5 | 27.5 | 22.2 | 23.8 | 28.3 |
| | $k_h$, $d^{-1}$ | 0.118 | 0.133 | 0.109 | - | - | - | - | - | - |
| | λ, d | - | - | - | 0.85 | 0.69 | 1.53 | 0.82 | 0.71 | 1.37 |
| 75 | $B_{max}$, L/kg DOM | 240 | 250 | 331 | 222 | 231 | 307 | 224 | 239 | 318 |
| | $R_m$, L/(kg DOM·d) | - | - | - | 25.7 | 27.2 | 32.3 | 26.7 | 28.2 | 34.1 |
| | $k_h$, $d^{-1}$ | 0.134 | 0.136 | 0.134 | - | - | - | - | - | - |
| | λ, d | - | - | - | 1.26 | 1.29 | 0.83 | 1.16 | 1.18 | 0.80 |
| 95 | $B_{max}$, L/kg DOM | 277 | 309 | 388 | 252 | 284 | 347 | 255 | 287 | 352 |
| | $R_m$, L/(kg DOM·d) | - | - | - | 25.8 | 33.7 | 39.7 | 26.9 | 34.5 | 40.7 |
| | $k_h$, $d^{-1}$ | 0.120 | 0.130 | 0.113 | - | - | - | - | - | - |
| | λ, d | - | - | - | 1.09 | 1.44 | 1.94 | 1.02 | 1.28 | 1.76 |
| 115 | $B_{max}$, L/kg DOM | 321 | 340 | 409 | 286 | 303 | 372 | 291 | 308 | 378 |
| | $R_m$, L/(kg DOM·d) | - | - | - | 25.2 | 26.7 | 35.3 | 26.6 | 28 | 37.2 |
| | $k_h$, $d^{-1}$ | 0.105 | 0.106 | 0.119 | - | - | - | - | - | - |
| | λ, d | - | - | - | 0.75 | 0.73 | 0.66 | 0.75 | 0.73 | 0.66 |
| 135 | $B_{max}$, L/kg DOM | 346 | 383 | 502 | 303 | 342 | 450 | 308 | 348 | 457 |
| | $R_m$, L/(kg DOM·d) | - | - | - | 27.8 | 30.3 | 39.9 | 28.8 | 31.7 | 41.7 |
| | $k_h$, $d^{-1}$ | 0.098 | 0.106 | 0.107 | - | - | - | - | - | - |
| | λ, d | - | - | - | 1.34 | 0.88 | 0.83 | 1.22 | 0.82 | 0.77 |
| 155 | $B_{max}$, L/kg DOM | 350 | 392 | 592 | 306 | 350 | 514 | 311 | 356 | 525 |
| | $R_m$, L/(kg DOM·d) | - | - | - | 29.0 | 34.0 | 39.4 | 30.0 | 35.3 | 41.2 |
| | $k_h$, $d^{-1}$ | 0.100 | 0.109 | 0.090 | - | - | - | - | - | - |
| | λ, d | - | - | - | 1.41 | 1.18 | 0.66 | 1.3 | 1.09 | 0.60 |
| 175 | $B_{max}$, L/kg DOM | 334 | 429 | 622 | 289 | 384 | 544 | 293 | 390 | 557 |
| | $R_m$, L/(kg DOM·d) | - | - | - | 27.7 | 35.2 | 39.4 | 28.5 | 36.6 | 41.4 |
| | $k_h$, $d^{-1}$ | 0.094 | 0.108 | 0.091 | - | - | - | - | - | - |
| | λ, d | - | - | - | 1.77 | 0.96 | 0.20 | 1.60 | 0.88 | 0.20 |

An analysis of the impact of process parameters of thermal disintegration on the value of estimated methane fermentation parameters was performed on the basis of the Gompertz model, which provided the best match between the calculated total biogas yield and the experimental data.

The rate of biogas production for raw bio-waste amounted to 26.8 L/(kg DOM·d). The determined maximum biogas yield for all thermally disintegrated bio-waste samples was higher than for the raw substrate. The highest value was obtained for bio-waste disintegrated within 2 h at 175 °C. Preliminary treatment of bio-waste under these conditions resulted in an increase of the $B_{max}$ parameter by 170%. Meanwhile, the maximum rate of biogas production $R_m$ (41 L/(kg DOM·d)) was obtained for samples disintegrated over a period of 2 h in temperatures ranging from 135–175 °C. The increase in the temperature of bio-waste pre-treatment to over 135 °C improved the maximum biogas yield for a disintegration time of 2 h. When the duration of the process was shorter, the increase in biogas production was insignificant.

In the methane fermentation process of thermally disintegrated bio-waste, the value of the parameter λ obtained in the Gompertz model depended on the conditions of the disintegration process. For the anaerobic stabilization of untreated bio-waste, the value of λ reached 1.3 d (32 h). The liquefaction of bio-waste at low temperatures either did not affect the value of this parameter or led to its increase. In the case of high-temperature disintegration—above 115 °C—the extension of the processing time resulted in a reduction of the duration of the lag phase for each sample. The lowest

value of this parameter, 0.2 d (5 h), was obtained for samples disintegrated at 175 °C over a period of 2 h.

Bearing in mind that the results of studies on thermal disintegration and methane fermentation of municipal waste published in the literature are few and far between, while those that are available mainly concern waste of very diverse morphology and chemical composition, it is difficult to compare the results obtained for municipal bio-waste with other experimental data. The values of estimated parameters for bio-waste and data provided by other authors are presented in Table 6.

**Table 6.** Comparison of the results of estimated parameters for bio-waste with the data of other authors (as cited by Syaichurrozi and Sumardiono [36]).

| Substrate | | $B_{max}$, L/kg DOM | $R_m$, L/(kg DOM.·d) | $\lambda$, $d^{-1}$ | Author |
|---|---|---|---|---|---|
| Vinasses (COD/N=600/7) | | 140.1 | 16.0 | 0.21 | Syaichurrozi and Sumardiono [36] |
| Cattle manure | | 418.3 | 9.5 | 4.46 | Budiyno et al. [28] |
| Municipal waste | | 522.0 | 97.0 | 1.20 | Zhu et al. [27] |
| Chicken manure | | 390.4 | 16.5 | 8.75 | Adiga et al. [37] |
| Raw bio-waste | | 206.2 | 26.8 | 1.30 | |
| Thermally disintegrated bio-waste | 0.5 h, 55 °C | 220.0 | 22.2 | 0.82 | |
| | 0.5 h, 175 °C | 293.0 | 28.5 | 1.60 | |
| | 1 h, 55 °C | 218.0 | 23.8 | 0.71 | By author |
| | 1 h, 175 °C | 390.0 | 36.6 | 0.88 | |
| | 2h, 55 °C | 273.0 | 28.3 | 1.37 | |
| | 2h, 175 °C | 557.0 | 41.4 | 0.20 | |

The value of the constant $k_h$ determined in the first-order model amounted to 0.14 $d^{-1}$ for bio-waste. According to other authors, $k_h$ can have the following values: 0.06–0.24 $d^{-1}$ for bio-waste [38] and 0.07–0.26 $d^{-1}$ for kitchen waste [38]

After the disintegration process, the value of constant hydrolysis in each case was lower than for the raw substrate. No significant correlation was found between the value of this parameter and process variables or the intensity of biogas production. Similar observations were made by Donoso-Bravo et al. [31], Ma et al. [39], and Polizzi et al. [40] in their studies, who also obtained lower $k_h$ values for disintegrated anaerobically-stabilized waste in comparison with raw substrates.

A constant $k_h$ does not explicitly indicate a trend of changes in the speed of hydrolysis in relation to the preliminary treatment of the substrate. As Polizzi et al. [40] show, when applying the first-order kinetic equation to describe the hydrolysis phase, the biological nature of enzymatic hydrolysis is not taken into account, disregarding the concentration of biomass, or rather the relationship between the concentration of substrate and that of biomass. Moreover, the values of the constant rate of hydrolysis of disintegrated substrates given in the literature differ significantly, because their value is influenced by experimental conditions (e.g., temperature, time, biomass adaptation) and substrate properties (e.g., solid matter content, particle size). Additionally, the effects of disintegration and biochemical hydrolysis are combined during pre-treatment as a cumulative outcome of various processes occurring during the liquefaction of the solid fraction of substrates [41].

Therefore, it can be stated that the hydrolysis constant $k_h$ is not an unequivocal indicator of the influence of thermal disintegration of substrates on the course of methane fermentation, although it is an important parameter in the complex modeling of the anaerobic decomposition process.

## 6. Conclusions

The conducted research has shown that, for each processing time, an increase in temperature was coupled with a higher degree of liquefaction of the solid fraction of bio-waste. The amount of biogas obtained in the methane fermentation of thermally disintegrated bio-waste and the share of methane in the biogas depended on the treatment conditions of disintegration. The highest total biogas

yield, after disintegration within 0.5 h, was obtained from samples subjected to treatment at 155 °C. In the case of the disintegration of the substrate over a period of 1 h, the total biogas yield grew with the increase in processing temperature (175 °C, 389 L/kg DOM, methane share: 68%). Processing at 175 °C for 2 h ensured the highest biogas yield increase compared to the raw substrate potential, amounting to 167% for bio-waste (methane share 63%). The disintegration process can, therefore, be a good solution to increase the energy efficiency of the methane fermentation of bio-waste by increasing biogas production, while reducing the amount of residual waste. Mathematical modeling can better optimize this process. The best match in the conducted research between the experimental data of biogas production in mesophilic methane fermentation from thermally disintegrated substrates was obtained for the Gompertz model.

Further research into the considered issues should include the development of modeling of thermal disintegration and processes occurring during the methane fermentation of liquefied substrates, taking into account the possibility of applying these models in the design and operational practice of bio-waste processing installations.

**Funding:** This research received no external funding

**Conflicts of Interest:** The author declares no conflict of interest.

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
