# Peer review of "Biogas and Methane Potential of Pre-Thermally Disintegrated Bio-Waste"

_energies, doi:10.3390/en12203880_

Round 1

Reviewer 1 Report

The study presents the research results on ‘Biogas and methane potential of bio-waste’. This topic is interesting; however, I cannot observe the advancement of the present study over the existing literature works and this manuscript cannot be acceptable in its present form.

There are many research results about biogas production in the view of pretreatment. What is the main advantages of this research? Please provide the advantage of these processes and compare to other recent research papers regarding the economic efficiency. What is the novelty of this research?

The explanation of results and discussion are not enough. Overall, this paper lacks the actual discussion of experimental data. They simply described their data. A more in-depth discussion other than description is needed.

Please improve the resolution in Figure 2 (including equations) and modify it to make it easier for the reader to understand.

It is not easy to understand the results of ‘Biogas and methane potential’ because the expression and discussion of the results in ‘Table 4’ are not sufficient. A detailed interpretation of the results and the addition of discussions is required.

Please compare to other recent research papers about yield (conversion) and productivity in the view of cost effectiveness and provide the explanations about overcoming the typical problem for using primary sludge in detail in the view of process efficiency.

The references are not adequately presented. Check recent relevant papers. I can see a few papers from 2017 to 2019 which are referred to mainly in the literature review.

Author Response

Thank you very much for spending Your valuable time and comments.

Below are the answers and the scope of the changes made in the manuscript. The language correction is marked in red in the manuscript.

There are many research results about biogas production in the view of pretreatment. What is the main advantages of this research? Please provide the advantage of these processes and compare to other recent research papers regarding the economic efficiency. What is the novelty of this research?

I agree that there is a lot of research in recent times regarding the pre-treatment processes of substrates before methane fermentation. However, they concern mostly of sewage sludge. The novelty of the conducted research, described in the article, which was marked in the first chapter (line 90-95) is to supplement the knowledge on thermal disintegration of bio-fraction separated from municipal waste, due to the lack of research data on this subject in the literature. Describing in the theoretical part the advantages of all methods of disintegration: chemical, thermal, mechanical, biological and mixed methods is difficult in a research article. A large amount of information in this part will change the form into a review article, which was not the subject of the work. However, the purpose of the research was to describe the kinetics of the fermentation process and the impact of parameters of disintegration on technological fermentation efficiency. No economic analysis was carried out, which may be the subject of separate studies and requires the use of multiple indicators. It would be difficult and very wide to include in one paper information on the methodology of kinetics and the methodology of economic analysis.

The explanation of results and discussion are not enough. Overall, this paper lacks the actual discussion of experimental data. They simply described their data. A more in-depth discussion other than description is needed.

Chapter 5 - the Discussion contains discussion of parameter values obtained in the models on the basis of test results (Chapter 4). The obtained values were compared with the literature data of other researchers (Table 6 and line 284-305). The most important achievement of the discussion was confirmation that it was shown, as in an only few researchers, that, it can be stated that the hydrolysis constant kh is not an unequivocal indicator of the influence of thermal disintegration of substrates on the course of methane fermentation, although it is an important parameter in complex modeling of the anaerobic decomposition process. ''

Please improve the resolution in Figure 2 (including equations) and modify it to make it easier for the reader to understand.

Figure 2 quality has been improved, axis descriptions have been added in charts.

It is not easy to understand the results of ‘Biogas and methane potential’ because the expression and discussion of the results in ‘Table 4’ are not sufficient. A detailed interpretation of the results and the addition of discussions is required.

The basis for the technological evaluation of the fermentation process are the parameters: the total biogas and methane production capacity per kg of dry matter and kg of organic dry matter, and share of methane (%), and these data were presented in Table 4. The values of these parameters are given for raw and disintegrated waste samples. The influence of process parameters of disintegration time and temperature on the values and changes of these parameters was commented. The test results presented in Table 4 were the basis for the modeling of kinetics and were discussed in Chapter 5 - Discussion. Based on the obtained real results in laboratory tests (Table 4), mathematical simulations were carried out using three models to obtain parameters determining the kinetics of the fermentation process for raw and disintegrated waste: maximum Bmax biogas production, maximum rate of biogas production Rm, parameter λ (lag phase), first-order kinetic constant k.

Please compare to other recent research papers about yield (conversion) and productivity in the view of cost effectiveness and provide the explanations about overcoming the typical problem for using primary sludge in detail in the view of process efficiency.

The research described in the article did not concern preliminary sewage sludge, only bio-fraction of municipal waste. Comparing the results of the research presented in the article with data on the efficiency and cost-effectiveness of sludge disintegration (a completely different substrate) was not the subject of the work. Adding content on sewage sludge will force the extension of the scope of introduction, a broader literature review. Such a comparison could be the subject of a review article on generally used disintegration methods, including thermal ones.

The references are not adequately presented. Check recent relevant papers. I can see a few papers from 2017 to 2019 which are referred to mainly in the literature review. 

 The literature has been completed.

Reviewer 2 Report

In general the introduction section is good and easy to follow, but statement of the motivation of the study is not intriguing.

I was hoping that some more real conclusions and direction could be made in conclusion. Rather than summarizing the research paper, a clear vision can or should be presented, and a conclusion is a right place for this.

The authors should clearly mention the future work in the conclusion section

Author Response

Thank you very much for spending Your valuable time and comments.

Below is the answer and the scope of the changes. The language correction is marked in red in the manuscript.

I was hoping that some more real conclusions and direction could be made in conclusion. Rather than summarizing the research paper, a clear vision can or should be presented, and a conclusion is a right place for this. The authors should clearly mention the future work in the conclusion section.

The conclusions have been changed.

Round 2

Reviewer 1 Report

Authors have provided reasonable response to this reviewer's comments.